# Stereospecific *syn*-dihalogenations and regiodivergent *syn*-interhalogenation of alkenes via vicinal double electrophilic activation strategy

Hyeon Moon [1,2], Jungi Jung [1,2], Jun-Ho Choi [1] ✉ & Won-jin Chung [1] ✉

Whereas the conventional *anti*-dihalogenation of alkenes is a valuable synthetic tool with highly predictable stereospecificity, the restricted reaction mechanism makes it challenging to alter the diastereochemical course into the complementary *syn*-dihalogenation process. Only a few notable achievements were made recently by inverting one of the stereocenters after *anti*-addition using a carefully designed reagent system. Here, we report a conceptually distinctive strategy for the simultaneous double electrophilic activation of the two alkene carbons from the same side. Then, the resulting vicinal leaving groups can be displaced iteratively by nucleophilic halides to complete the *syn*-dihalogenation. For this purpose, thianthrenium dication is employed, and all possible combinations of chlorine and bromine are added onto internal alkenes successfully, particularly resulting in the *syn*-dibromination and the regiodivergent *syn*-bromochlorination.

Traditional oxidative 1,2-dihalogenation of alkenes with common electrophilic reagents such as elemental dihalogens ($Cl_2$, $Br_2$) generally proceeds via the intermediacy of a cationic cyclic species (**1**), a haliranium ion, resulting in stereospecific *anti*-addition (**2**) upon the subsequent stereo-invertive $S_N2$ displacement by halide from the opposite side (Fig. 1a)[1]. Such transformation is one of the oldest and most well-established reactions in organic chemistry and thus has been widely utilised as a straightforward synthetic tool to access vicinal dihalides from geometrically defined alkenes with predictable stereospecificity[2]. However, because of the highly restricted reaction mechanism, it has been nearly impossible to alter the stereochemical course. Consequently, the complementary *syn*-dihalogenation process is rare[3] and often approached via a multi-step operation[4-7]. For example, *cis*−1,2-dichlorocyclohexane is produced through a sequence relying on an available *syn*-stereospecific alkene activation such as epoxidation followed by deoxydichlorination[8], which indicates that *syn*-dichlorination methods are underdeveloped. Moreover, the relative configuration of vicinal dihalide products is dependent on the geometry of the starting alkenes, and thus the synthetic utility of dihalogenations has often been limited by the accessibility of the desired alkene geometry because only one diastereochemical course has been available (Fig. 1b). During the study on conformational and configurational analysis of chlorinated natural products by the Carreira group[9], the preparation of *anti*-dichloride (**3**) was easily accomplished by the straightforward single-step dichlorination of the naturally occurring *E*-isomer of ethyl sorbate (**4**) with $Et_4NCl_3$. On the other hand, the corresponding *syn*-dichloride (**5**) had to be synthesised by a cumbersome three-step operation involving alkene epoxidation and two iterative deoxychlorinations because the *Z*-isomer of ethyl sorbate was not readily available. *syn*-Dihalogenation would also be useful when the alkene reactant is produced as a separable mixture of geometrical isomers. For instance, the alkene intermediates **6** were obtained with poor *E/Z* selectivity via the Wittig or the Julia olefinations during the syntheses of chlorosulfolipids by the Vanderwal group[10-14] and the Carreira group[15-17], and only (*Z*)-**6** could be further utilised for the subsequent *anti*-dichlorination step. The presence of a *syn*-addition

[1]Department of Chemistry, Gwangju Institute of Science and Technology, Gwangju 61005, Republic of Korea. [2]These authors contributed equally: Hyeon Moon, Jungi Jung. ✉e-mail: junhochoi@gist.ac.kr; wjchung@gist.ac.kr

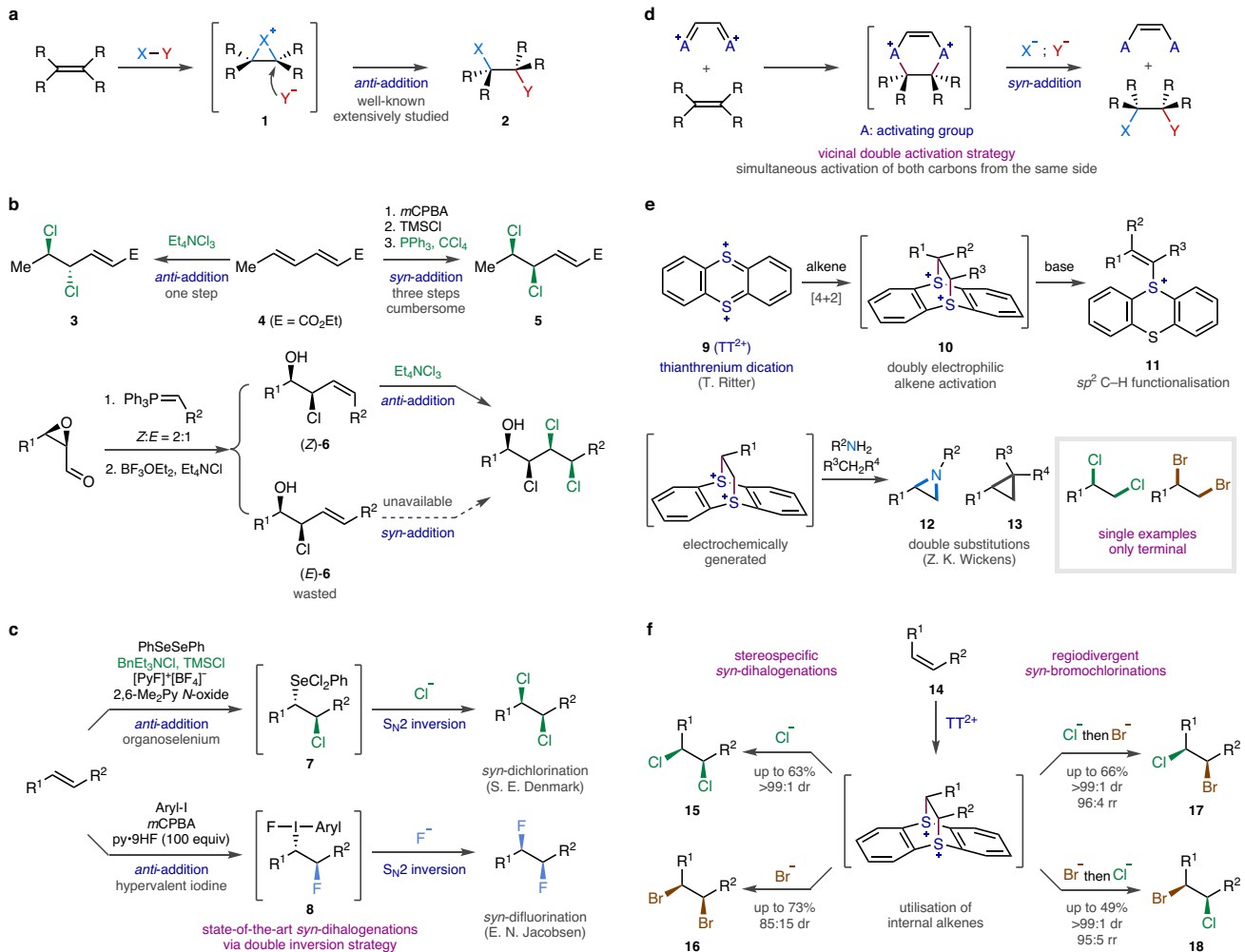

**Fig. 1 | Research outline. a** Traditional *anti*-dihalogenation. **b** Needs for complementary *syn*-dihalogenation. **c** Current state-of-the-art via double inversion strategy. **d** Our approach of vicinal double activation strategy. **e** Thianthrenium dication as doubly electrophilic alkene activator and its usage. **f** This work: *syn*-stereospecific and regiodivergent dihalo- and interhalogenations.

method would have enabled the transformation of (*E*)-**6** into the same desired dichlorination product, thereby increasing the overall efficiency of the synthesis. Additionally, instead of relying on poorly *Z*-selective Wittig olefination in the previous step, it would have been possible to consider highly *E*-selective olefination, which was unfortunately not a viable option. Clearly, it is highly desirable to develop diverse *syn*-stereospecific alternatives, which will greatly simplify the current multi-step synthetic route and provide invaluable synthetic tools for organic chemists.

In the past decade, whereas the conventional *anti*-dihalogenation chemistry has advanced remarkably, as highlighted by the realisations of enantioselective variants[18–25], the stereochemically opposite *syn*-dihalogenation has received much less attention. Only recently, a few notable accomplishments have been reported (Fig. 1c). In 2015, the Denmark group developed the first example of generally applicable, one-step *syn*-dichlorination of alkenes by taking advantage of organoselenium's susceptibility to oxidation[26,27]. After the well-established *anti*-addition of phenylselenyl chloride across a C=C bond, the facile in-situ oxidation of the selenium(II) species leads to the formation of a highly electron-deficient selenium(IV) leaving group (**7**), allowing the subsequent $S_N2$ displacement by chloride to result in overall *syn*-addition. Shortly after, in 2016, diastereoselective 1,2-difluorination of alkenes including *syn*-stereospecific examples was developed by the Jacobsen group with hypervalent iodine(III) difluoride[28]. This reaction proceeds in a similar fashion via *anti*-iodofluorination (**8**) followed by

stereo-invertive nucleophilic substitution of iodine(III) with fluoride. Although these two pioneering examples are remarkable achievements, there still are drawbacks. The reaction conditions involve either a complex combination of several reagents or a large excess of toxic substance. In addition, E2 elimination is often accompanied as a competing side reaction. Moreover, *syn*-dibromination remains elusive[29]. Furthermore, scope expansion with two different halogen species would be an even harder challenge, which cannot be addressed by the currently available approaches. Therefore, this important research field is still at a very early stage of development, and the invention of a mechanistically distinctive *syn*-dihalogenation is highly desired.

To that end, we devised a synthetic strategy, in which the two carbons of alkene are simultaneously activated from the same side with two electron-deficient moieties via a stereospecific transformation such as cycloaddition (Fig. 1d). Then, each electrophilic carbon can be displaced by halide to give the *syn*-dihalogenated product. Whereas one leaving group is shared by both carbons in the traditional alkene dihalogenation, allowing only one substitution, each carbon has its own leaving group in our case, and thus two displacements can proceed to alter the diastereochemical outcome. A reagent with such a doubly electrophilic property has been recently developed by the Ritter group (Fig. 1e)[30,31]. It was reported that thianthrenium dication (**9**, $TT^{2+}$) could be utilised for the stereospecific construction of two vicinal C−S⁺ bonds on the same side of alkenes simultaneously through

a hetero Diels–Alder reaction. However, this species has rarely been employed for substitution chemistry. In most cases, the bissulfonium intermediate (**10**) was treated with a base to form *S*-alkenyl thianthrenium cation (**11**), which was then manipulated in various ways, resulting in overall sp[2] C–H functionalisation. Furthermore, although the cycloadducts have been characterised with internal alkenes such as (*E*)- and (*Z*)−4-octenes[30], the typical substrate scope includes mostly terminal alkenes and a few cyclic ones. The most notable examples of double substitutions were disclosed recently by the Wickens group for the efficient synthesis of *N*-alkyl aziridines (**12**)[32] and cyclopropanes (**13**)[33] as well as for regiospecific 1,2-aminofunctionalisations[34]. In their study, a cationic pool of the thianthrenium-activated species was generated via a carefully modulated electrochemical oxidation and subsequently reacted with various nitrogen or carbon nucleophiles[35]. Then, as a scope expansion, other nucleophiles including halides were also examined, and the feasibility of 1,2-dihalogenation was demonstrated[32]. However, only a single example for each halogen (Cl and Br) was described. More critically, only terminal alkenes could be employed in their reaction system, and thus complex stereochemical aspects such as stereospecificity could not be addressed. Nonetheless, on the basis of these precedents, we saw a promising potential from TT[2+] as it possesses the desired properties for our reaction concept of stereospecific vicinal double electrophilic activation of alkenes.

Herein, we describe *syn*-stereospecific and regiodivergent installations of all possible combinations of chlorine and bromine onto alkenes utilising TT[2+] and organic-soluble halides (Fig. 1f). Remarkably, it is worth noting that *syn*-dibromination (**16**) and *syn*-bromochlorinations (**17** and **18**) are accomplished through this approach. Furthermore, the evidence for direct stereospecific substitutions of the dicationic alkene-TT[2+] adduct is provided through mechanistic studies.

## Results and discussion
### Stereospecific *syn*-dihalogenations
Our initial reaction condition screening was carried out with a disubstituted (*Z*)-alkene **14a** (1.0 equiv), thianthrene-*S*-oxide (**19**, TTO, 1.0 equiv), triflic anhydride (Tf$_2$O, 1.0 equiv), and tetra-*n*-butylammonium chloride (3.0 equiv) in CH$_2$Cl$_2$ at 0 °C (Table 1). After the formation of the cycloadduct **10**, S$_N$2 substitutions of the sulfoniums by two chlorides afforded the desired *syn*-dichlorinated product **15a** in 61% yield with complete diastereoselectivity (entry 1). When a smaller amount of chloride was used, the product yield was decreased because of the incomplete conversion (entry 2). The use of an excess amount of *n*-Bu$_4$NCl led to a slightly improved yield, but the operation became cumbersome (entry 3). No meaningful change was observed upon dilution or concentration (entries 4 and 5). Moreover, the chlorination step at different temperatures resulted in attenuated yield and/or diastereoselectivity (entries 6 and 7). In addition, the examination of another quaternary ammonium countercation, Me$_3$PhNCl as a chloride source provided a lower yield (entry 8). Furthermore, an extensive survey of solvents was conducted, but inferior conversions were obtained mostly because of the poor solubility of the ionic intermediate **10** and/or halide reagent (See Section 2.6 of the Supplementary Information).

With the optimal reaction conditions in hand, the scope of *syn*-dihalogenation was evaluated (Fig. 2). When the benzoate group was located in closer proximity to the alkene compared to **15a**, a reasonably high diastereoselectivity was obtained with a similar yield (**15b**). The marginal erosion of stereospecificity may have been caused by the competitive anchimeric assistance of the pendant nucleophile[36]. A slightly bulkier ethyl group was well-tolerated, providing a good yield and an excellent diastereoselectivity (**15c**). The impact of steric hindrance became apparent with the introduction of a β-branched isobutyl group, leading to a substantially attenuated yield but with a

### Table 1 | Reaction condition optimisation for *syn*-dichlorination[a]

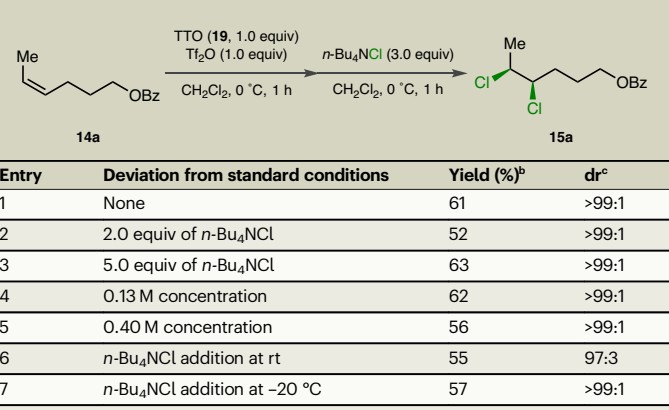

| Entry | Deviation from standard conditions | Yield (%)[b] | dr[c] |
|---|---|---|---|
| 1 | None | 61 | >99:1 |
| 2 | 2.0 equiv of *n*-Bu$_4$NCl | 52 | >99:1 |
| 3 | 5.0 equiv of *n*-Bu$_4$NCl | 63 | >99:1 |
| 4 | 0.13 M concentration | 62 | >99:1 |
| 5 | 0.40 M concentration | 56 | >99:1 |
| 6 | *n*-Bu$_4$NCl addition at rt | 55 | 97:3 |
| 7 | *n*-Bu$_4$NCl addition at −20 °C | 57 | >99:1 |
| 8 | Me$_3$PhNCl as chloride source | 44 | >99:1 |

*Bz* benzoyl, *TTO* thianthrene-*S*-oxide, *Tf* trifluoromethanesulfonyl, *dr* diastereomeric ratio.
[a]Standard conditions: 1.0 mmol scale at 0.25 M concentration.
[b]Isolated yields after column chromatography.
[c]Determined by [1]H NMR analysis of the purified materials.

still excellent diastereoselectivity (**15d**). In the absence of nearby branching, long primary alkyl chains were amenable on both sides of alkene (**15e**). A few heteroatom-containing moieties such as thiophene (**15f**), benzodioxole (**15g**), and phthalimide (**15h**) were also compatible. Unfortunately, a tethered alkyl ether function appeared to cause the decomposition of the bissulfonium intermediate. Nonetheless, by conducting the chlorination immediately, one minute after thianthrenation, the reactive cycloadduct could be preserved in some degree, and the desired product was obtained with moderate efficiency (**15i**). Such interference was marginally alleviated by employing a bulky silyl ether, providing **15j** in an improved yield. However, aryl ether was even more problematic because TT[2+] (**9**) was intercepted by electron-rich arene, and thus the corresponding dichloride was not detected at all (not shown), being consistent with the well-established aryl C–H activating property of **9**[37–42]. In the presence of two electronically distinct alkenes, chemoselectivity was observed as only the relatively electron-rich alkene underwent *syn*-dichlorination (**15k**). Subsequently, the halogen scope was expanded to *syn*-dibromination, which has never been realised before. Gratifyingly, **16a** was obtained by employing tetra-*n*-butylammonium bromide in a high yield with an appreciable level of 83:17 diastereoselectivity, constituting a significant example of *syn*-dibromination. The slight loss of stereospecificity compared to the chlorination counterpart (**15a**) was probably caused by the neighbouring group participation of the first-installed bromine substituent, which would promote the *anti*-addition pathway involving a cyclic bromiranium species[43–47]. From the examination of a few other substrates, the corresponding *syn*-dibrominated products (**16f**, **16h**, and **16m**) were successfully afforded with synthetically useful yields and diastereoselectivities. Subsequently, the influence of alkene geometry was evaluated. To our delight, a complete diastereoselectivity was obtained from *syn*-dichlorination of (*E*)-alkene (**15l**) although a prolonged reaction time was required to achieve a reasonable yield. In the case of *syn*-dibromination, the high reactivity was maintained, but the portion of the *syn*-addition isomer was decreased (**16l**) probably because the bromiranium formation became geometrically favourable enough to outcompete the second bromide attack. On the other hand, the attempts with iodide and fluoride nucleophiles did not provide the 1,2-dihalides under similar reaction conditions.

**Fig. 2 | Substrate scope of *syn*-dichlorination and *syn*-dibromination.** [a]All reactions were performed on 1.0 mmol scale at 0.25 M concentration. Isolated yields after column chromatography are given. Diastereomeric ratios (dr) were determined by [1]H NMR analysis of the purified materials. [b]Chlorination at −40 °C. [c]With 1.2 equiv of TTO and Tf₂O. [d]Thianthrenation for 1 minute. [e]With 1.1 equiv of Tf₂O. [f]Chlorination for 5 hours. (Phth = phthaloyl, TBDPS = *tert*-butyldiphenylsilyl).

## Regiodivergent *syn*-bromochlorination

During the optimisation of the reaction conditions, it was noticed that the first nucleophilic substitution occurred significantly faster than the second substitution. Furthermore, it was observed that the first halide addition exhibited high site-selectivity, leading to the formation of a monohalogenated TT⁺ intermediate with excellent isomeric purity (Fig. 3). Therefore, it was hypothesised that the second halogenation could be performed with a different halide. Indeed, the sequential treatment with chloride and then bromide resulted in a *syn*-inter-halogenation. Although site-selective difunctionalisation of terminal alkene-TT²⁺ adduct was reported before[34], such high positional discrimination with internal alkenes had never been realised prior to our work. Even more remarkably, the regiochemistry could be inverted simply by reversing the addition order, allowing access to both constitutional isomers 17 and 18.

For the successful operation of these site-selective inter-halogenations, the first halogenation must be completed without overhalogenation. The premature addition of the second halide before the full consumption of the bissulfonium species would complicate the reaction outcome via unordered halogen combinations such as homo-dihalogenation and halogen scrambling. Therefore, the first step was conducted at a low temperature with strictly one equivalent of the halide reagent for an extended period of time. Even though the reaction is substantially slowed down, especially with less nucleophilic chloride, an excess amount should not be used. Furthermore, in some cases, the ionic reaction intermediates became insufficiently soluble under the modified conditions, which necessitated the addition of a polar solvent such as *N,N*-dimethylformamide (DMF) to compensate for the decreased solubility and thereby increase the reaction rate.

The scope of regiodivergent *syn*-bromochlorination was examined with several (*Z*)-alkenes 14 under slightly adjusted conditions for each substrate (Fig. 4) (See Sections 2.7 and 2.8 of the Supplementary Information). Although small amounts of homo-dihalides and constitutional isomers were formed as inseparable side products, the desired *syn*-bromochlorides 17 and 18 were successfully obtained as the major components with high site-selectivity and excellent

**Fig. 3 | Regiodivergent *syn*-Interhalogenation.** Site-selective sequential additions of two different halides.

diastereoselectivity. Again, benzoates and phthalimide at variable proximity were tolerated. Notably, when the electron-withdrawing group was closer to the reacting alkene, the site-selectivity was improved (17a/18a vs 17b/18b). The efficiency was somewhat attenuated when bromide was added first (18) probably because of the internal nucleophilic participation of the bromine substituent as the formation of a transient bromiranium intermediate may have scrambled the position of the second substitution. Nonetheless, such high levels of controllable regiochemical discretion have never been realised even for the more studied *anti*-bromochlorination[19,22,24,47]. The site-selectivity and *syn*-stereospecificity of both bromochlorinations were unambiguously established by the X-ray crystallographic analysis of 17n and 18n (See Section 3.1 of the Supplementary Information).

## Mechanistic studies

Through the inspection of the experimental results from the bromo-chlorinations, a few mechanistic aspects regarding selectivities are discussed (Fig. 5). At first glance, the site-selectivity appears to be influenced to some extent by the electronic difference between the two alkyl substituents because the selectivity decreases as the electron-withdrawing benzoate group moves away from the reacting alkene (Fig. 5a, 17a vs 17b). To test this hypothesis, a substrate with a much longer alkyl tether was examined so that the electronic bias would become negligible. However, only a marginally altered, still comparably high site-selectivity was obtained (17q), suggesting that the electronic effect is a minor determining factor. Instead, the steric difference is more likely to be responsible for the discriminated attack

**Fig. 4 | Substrate scope of regiodivergent *syn*-bromochlorination.** [a]All reactions were performed on 1.0 mmol scale at 0.25 M concentration. Yields of the isolated materials after column chromatography are given, and the calculated yields of the interhalides are shown in the parentheses. Diastereomeric ratios (dr) and regioisomeric ratios (rr) were determined by [1]H NMR analysis of the purified materials. [b]With 0.3 mL of DMF. [c]With 0.5 mL of DMF.

of nucleophile[48]. To account for the observed site-selectivity, the first halogenation must take place preferably at the more substituted methylene side. The nucleophilic chlorination of a bissulfonium species ($14 + TT^{2+}$) was analysed as a representative process via the density functional theory calculation using the GAMESS(US) 2018 software package[49–51] at the $w$B97X-D/6-31 + G(d,p)/PCM(CH$_2$Cl$_2$) level of theory (Fig. 5b, Supplementary Data 1)[52–56]. The computed activation energy difference between the two isomeric pathways is ca. 1.5 kcal/mol favouring the major constitutional isomer, which appears to be promoted by relieving the steric congestion in the more crowded side of the alkene-TT$^{2+}$ adduct. This argument is supported by an NCI plot analysis[57,58] that allows the comparison between methyl and ethyl groups. Whereas the methyl group does not engage in any apparent interactions, the ethyl group experiences repulsive forces with the thianthrene moiety (See Section 3.2 of the Supplementary Information). Furthermore, a plausible reaction pathway was proposed on the basis of the observed excellent stereospecificity (Fig. 5c). Alkenyl TT$^+$ species has been identified as a key intermediate from the studies by the Wickens group on their double nucleophilic functionalisations of terminal alkenes under basic conditions[33]. In addition, the direct transformations of isolated alkenyl TT$^+$ into various cyclic compounds have been shown by the Ritter group[31] and the Shu group[59]. However, such an elimination process is likely to cause a loss of stereochemical information in our system that employs internal alkenes. For the subsequent hydrohalogenation to be stereospecific, the protonation must take place both selectively from the same side with halide and rapidly prior to the C−C bond rotation, which are quite demanding requirements to meet. Instead, our *syn*-dihalogenations under neutral conditions are likely to proceed via two sequential S$_N$2 displacements without involving an alkenyl TT$^+$ species because excellent stereospecificity was observed from both alkene geometrical isomers. To support our mechanistic hypothesis regarding the reactive intermediate, a control experiment was performed (Fig. 5d). An alkenyl TT$^+$ (14a-TT$^+$) was prepared separately[30] and subjected to the standard

reaction of a different alkene substrate (**14g**). As expected, no dihalogenated product (**15a**) derived from alkenyl TT$^+$ was observed while the *syn*-dichlorination of alkene proceeded uneventfully to give **15g**, confirming that alkenyl TT$^+$ is not an active species in our reaction.

In conclusion, we have developed a distinctive synthetic method for the *syn*-stereospecific alkene dihalogenations through the simultaneous activation of C−C double bonds from the same side. To realize this electrophilic vicinal double activation strategy, thianthrenium dication was successfully utilised for the concerted installation of vicinal double leaving groups on internal alkenes. Then, the iterative nucleophilic displacements with chloride and/or bromide enabled *syn*-additions of all possible halogen combinations with excellent diastereoselectivities. In particular, *syn*-dibromination and regiodivergent *syn*-bromochlorinations were accomplished. Upon the analysis of the experimental data, the site-selectivity appeared to be dominated by steric effect, which was supported by DFT calculation, and the exquisite stereospecificity was attributed to the absence of the competing elimination pathway under our mild reaction conditions, avoiding the formation of commonly observed alkenyl thianthrenium cation. Our work provides a mechanistically unique approach toward *syn*-stereospecific alkene dihalogenations, and the scope of this underdeveloped transformation has been greatly expanded.

## Methods

### A representative procedure for *syn*-dichlorination of alkenes

To a stirred solution of alkene **14a** (204 mg, 1.00 mmol, 1.0 equiv) and thianthrene-*S*-oxide (**19**, 232 mg, 1.00 mmol, 1.0 equiv) in CH$_2$Cl$_2$ (4 mL) was added Tf$_2$O (168 µL, 1.00 mmol, 1.0 equiv) at 0 °C. After 1 hour, a solution of *n*-Bu$_4$NCl (834 mg, 3.00 mmol, 3.0 equiv) in CH$_2$Cl$_2$ (8 mL) was added. After 1 hour, H$_2$O (10 mL) was added, and the organic layer was separated. The aqueous layer was extracted with CH$_2$Cl$_2$ (10 mL × 3), and the combined organic extracts were dried over MgSO$_4$ (2 g), filtered through a glass frit, and concentrated in vacuo. The residue was diluted with Et$_2$O (5 mL), filtered through a pad of SiO$_2$

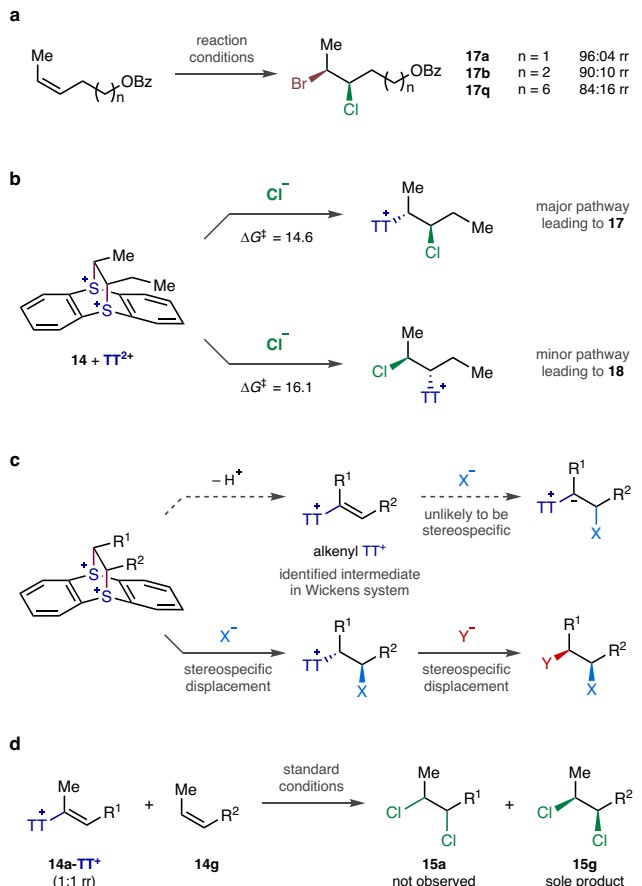

**Fig. 5 | Mechanistic discussion. a** Examination of the substituent effect on the site-selectivity of bromochlorination. **b** Computational analysis on the site-selectivity of initial chlorination ($w$B97X-D/6-31 + G(d,p)/PCM(CH$_2$Cl$_2$), free energies in kcal/mol). **c** Probable consequence of an alkenyl TT$^+$ intermediate and the likely pathway of our system. **d** Experimental validation of the reactive species.

($\phi$ = 2.0 cm, $l$ = 4.0 cm, Et$_2$O, 150 mL), and concentrated in vacuo. The crude material was purified by flash column chromatography (SiO$_2$, $\phi$ = 5.5 cm, $l$ = 15 cm, EtOAc/hexanes = 1/30, R$_f$ = 0.30) and Kugelrohr distillation (0.15 mmHg, 220 °C) to afford **15a** (169 mg, 61%, >99:1 dr) as a colourless oil. Data for **15a**[60]: $^1$H NMR (400 MHz, CDCl$_3$): δ 8.06–8.03 (m, 2H), 7.59–7.55 (m, 1H), 7.47–7.43 (m, 2H), 4.42–4.33 (m, 2H), 4.12 (appr. pent, $J$ = 6.6, 1H), 4.00 (ddd, $J$ = 9.2, 6.7, 2.8, 1H), 2.26–2.07 (m, 2H), 1.98–1.87 (m, 2H), 1.65 (d, $J$ = 6.4, 3H).

## A representative procedure for *syn*-bromochlorination of alkenes

To a stirred solution of alkene **14a** (204 mg, 1.00 mmol, 1.0 equiv) and thianthrene-*S*-oxide (**19**, 232 mg, 1.00 mmol, 1.0 equiv) in CH$_2$Cl$_2$ (4 mL) was added Tf$_2$O (168 μL, 1.00 mmol, 1.0 equiv) at 0 °C. After 1 hour, the reaction mixture was cooled to −30 °C, and a solution of $n$-Bu$_4$NCl (278 mg, 1.00 mmol, 1.0 equiv) in CH$_2$Cl$_2$ (2.5 mL) was added. After 24 hours, the reaction mixture was warmed to 0 °C, and a solution of $n$-Bu$_4$NBr (322 mg, 1.00 mmol, 1.0 equiv) in CH$_2$Cl$_2$ (2.5 mL) was added. After 1 hour, H$_2$O (10 mL) was added, and the organic layer was separated. The aqueous layer was extracted with CH$_2$Cl$_2$ (10 mL × 3), and the combined organic extracts were dried over MgSO$_4$ (2 g), filtered through a glass frit, and concentrated in vacuo. The residue was diluted with Et$_2$O (5 mL), filtered through a pad of SiO$_2$ ($\phi$ = 2.0 cm, $l$ = 4.0 cm, Et$_2$O, 150 mL), and concentrated in vacuo. The crude material was purified by flash column chromatography (SiO$_2$, $\phi$ = 4.0 cm, $l$ = 11 cm, CH$_2$Cl$_2$/hexanes = 1/3, R$_f$ = 0.30) and Kugelrohr

distillation (0.15 mmHg, 230 °C) to afford a colourless oil (220 mg, 69% (59% **17a**), 99:1 dr, 90:10 rr). Data for **17a**: $^1$H NMR (400 MHz, CDCl$_3$): δ 8.06–8.03 (m, 2H), 7.59–7.52 (m, 1H), 7.47–7.43 (m, 2H), 4.39–4.36 (m, 2H), 4.23–4.16 (m, 1H), 4.08–4.04 (m, 1H), 2.32–2.23 (m, 1H), 2.14–2.05 (m, 1H), 2.00–1.88 (m, 2H), 1.84 (d, $J$ = 6.7, 3H); $^{13}$C NMR (100 MHz, CDCl$_3$): δ 166.7, 133.1, 130.3, 129.7, 128.5, 66.9, 64.2, 52.1, 33.0, 25.7, 23.6; HRMS (ESI): [M + H]$^+$ calcd for C$_{13}$H$_{17}$$^{79}$Br$^{35}$ClO$_2$, [C$_{13}$H$_{17}$$^{81}$Br$^{35}$ClO$_2$ + C$_{13}$H$_{17}$$^{79}$Br$^{37}$ClO$_2$], C$_{13}$H$_{17}$$^{81}$Br$^{37}$ClO$_2$: 319.0095 (76.5%), 321.0073 (100.0%), 323.0050 (25.2%); found: 319.0108 (76.1%), 321.0086 (100.0%), 323.0059 (23.2%).

## Data availability
The data supporting the findings of this study are available within this article and its Supplementary Information, which includes experimental details, characterisation data, copies of NMR spectra for all new compounds, and DFT calculation data. The computation output files are provided as Supplementary Data 1. Crystallographic data for **17n** and **18n** have been deposited at the Cambridge Crystallographic Data Centre (CCDC), under deposition numbers CCDC 2262333 and 2262334. Copies of the data can be accessed free of charge via https://www.ccdc.cam.ac.uk/structures/. All data are available from the corresponding author upon request.

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

## Acknowledgements

This research was supported by the Korea Toray Science Foundation (W.J.C.). We thank the Surface Physical Property Lab at GIST Central Research Facilities (GCRF) for the X-ray crystallographic analysis of **17n** and **18n**.

## Author contributions

W.J.C. conceived the research concept. J.H.C. directed the computational study. W.J.C., H.M. and J.J. designed the synthetic strategy. H.M. and J.J. performed the synthetic work. W.J.C. and H.M. conducted the DFT calculation. W.J.C., H.M., and J.J. wrote the manuscript. All authors discussed the results and contributed to editing the manuscript and preparing the Supplementary Information.

## Competing interests

The authors declare no competing interests.
