## [Peer Review File · Nature Communications]

Editorial Note: Parts of this peer review file have been redacted as indicated to maintain the confidentiality of other journals.

REVIEWER COMMENTS

Reviewer #2 (Remarks to the Author):

I feel the author appropriately addressed the reviewer comments, including my own (I was reviewer 2 in the [redacted] submission). The toned down discussion of prior work and mechanistic explanations is now appropriate and places the work in a clear context. I agree entirely with the authors' argument that the prior related work from Ritter and Wickens is not the same as what is presented in this manuscript. This is particularly underscored by the fact that subsequent work from Wickens has revealed that alkenyl thianthrenium salts, rapidly formed by elimination of the dicationic adducts, are the active intermediate that engages nucleophiles rather than the dicationic adducts themselves. As a result, this is the first example wherein direct evidence of substitution of the dicationic adducts is supported (since syn-dihalogenation is not expected to occur from the alkenyl thianthrenium intermediate, as the authors state in the manuscript). The work shown in this manuscript has meaningful synthetic limitations, as all three reviewers noted in the initial review, but I think this work is nonetheless an important contribution to the rapidly expanding collection of researchers interested in both syn-dihalogenation and thianthrenium salts.

The way the authors describe why they favor direct substitution at the adducts is reasonable in the revision. That said, if I were the authors, I would add an experiment that validates their very reasonable explanation. Given that alkenyl thianthrenium salts are straightforward to prepare, the authors could illustrate that these species do not provide the same products as direct reaction of the dicationic adducts when treated halide salts. The alkenyl thianthrenium salts could be conclusively shown to either not react with halides or shown to give a distinct stereochemical outcome. Care would need to be placed to maintain similar reaction conditions since the elimination of the dicationic adducts would generate HX species. The cleanest way to test this would perhaps be to add an alkenylthianthrenium derived from a different alkene to a standard reaction and check if it also converts to the syn-dihalide or if only the dicationic adduct is converted. Regardless, these experiments are not strictly needed but rather I think they would enhance the overall manuscript. If

it turned out the alkenyl thianthrenium does furnish syn products, this would certainly be surprising and, while that outcome would require some rewriting, it would be quite interesting.

The authors could also consider citing a very recent review of thianthrenium salts as a tool in alkene functionalization that was published during the review process of this manuscript (<https://doi.org/10.1002/anie.202314904>) given the relevance to the topic at hand.

Overall, I fully support publication of this revised manuscript in Nat. Comm. but I would encourage the authors to use experiments to conclusively exclude the alkenyl thianthrenium intermediate unless there is some clear experimental reason this cannot be studied. Regardless, this is a great contribution to the field and I learned a lot from the discoveries the authors present in this manuscript.

Point-by-Point Response to Reviewer's Comments:

Reviewer #2 (Remarks to the Author):

I feel the author appropriately addressed the reviewer comments, including my own (I was reviewer 2 in the [redacted] submission). The toned down discussion of prior work and mechanistic explanations is now appropriate and places the work in a clear context. I agree entirely with the authors' argument that the prior related work from Ritter and Wickens is not the same as what is presented in this manuscript. This is particularly underscored by the fact that subsequent work from Wickens has revealed that alkenyl thianthrenium salts, rapidly formed by elimination of the dicationic adducts, are the active intermediate that engages nucleophiles rather than the dicationic adducts themselves. As a result, this is the first example wherein direct evidence of substitution of the dicationic adducts is supported (since syn-dihalogenation is not expected to occur from the alkenyl thianthrenium intermediate, as the authors state in the manuscript). The work shown in this manuscript has meaningful synthetic limitations, as all three reviewers noted in the initial review, but I think this work is nonetheless an important contribution to the rapidly expanding collection of researchers interested in both syn-dihalogenation and thianthrenium salts.

The way the authors describe why they favor direct substitution at the adducts is reasonable in the revision. That said, if I were the authors, I would add an experiment that validates their very reasonable explanation. Given that alkenyl thianthrenium salts are straightforward to prepare, the authors could illustrate that these species do not provide the same products as direct reaction of the dicationic adducts when treated halide salts. The alkenyl thianthrenium salts could be conclusively shown to either not react with halides or shown to give a distinct stereochemical outcome. Care would need to be placed to maintain similar reaction conditions since the elimination of the dicationic adducts would generate HX species. The cleanest way to test this would perhaps be to add an alkenylthianthrenium derived from a different alkene to a standard reaction and check if it also converts to the syn-dihalide or if only the dicationic adduct is converted. Regardless, these experiments are not strictly needed but rather I think they would enhance the overall manuscript. If it turned out the alkenyl thianthrenium does furnish syn products, this would certainly be surprising and, while that outcome would require some rewriting, it would be quite interesting.

: We have performed a control experiment following the reviewer's suggestion. An alkenyl TT^+ was prepared and subjected to a standard reaction of a different alkene. Then, no product was formed from the alkenyl TT^+ , confirming that it is not the active species in our system. This result has been included as Fig. 5d.

The authors could also consider citing a very recent review of thianthrenium salts as a tool in alkene functionalization that was published during the review process of this manuscript (<https://doi.org/10.1002/anie.202314904>) given the relevance to the topic at hand.

: The suggested reference has been included as ref. 35.

Overall, I fully support publication of this revised manuscript in Nat. Comm. but I would encourage the authors to use experiments to conclusively exclude the alkenyl thianthrenium intermediate unless there is

some clear experimental reason this cannot be studied. Regardless, this is a great contribution to the field and I learned a lot from the discoveries the authors present in this manuscript.